# Life Stressors in Patients with Inflammatory Bowel Disease: Comparison with a Population-Based Healthy Control Group in the Czech Republic

**DOI:** 10.3390/ijerph18063237

**Published:** 2021-03-21

**Authors:** Hana Bednarikova, Natalia Kascakova, Jana Furstova, Zuzana Zelinkova, Premysl Falt, Jozef Hasto, Peter Tavel

**Affiliations:** 1Olomouc University Social Health Institute, Palacky University, Univerzitni 22, 779 00 Olomouc, Czech Republic; natalia.kascakova@oushi.upol.cz (N.K.); jana.furstova@oushi.upol.cz (J.F.); zuzana.zelinkova@nsmas.sk (Z.Z.); j.hasto.tn@gmail.com (J.H.); peter.tavel@oushi.upol.cz (P.T.); 2Department of Gastroenterology and Digestive Endoscopy, University Hospital—St. Michael’s Hospital, Satinskeho 1, 811 08 Bratislava, Slovakia; 32nd Department of Internal Medicine, University Hospital Olomouc, I.P. Pavlova 185/6, 779 00 Olomouc, Czech Republic; premysl.falt@fnol.cz; 4Faculty of Medicine, Palacky University, Krizkovskeho 511/8, 779 00 Olomouc, Czech Republic; 5Psychiatric-Psychotherapeutic Outpatient Clinic, Pro Mente Sana, Heydukova 27, 811 06 Bratislava, Slovakia; 6Department of Social Work, St. Elizabeth College of Health and Social Work, Namestie 1. maja 1, 810 00 Bratislava, Slovakia; 7Faculty of Medicine, Department of Psychiatry, Slovak Medical University in Bratislava, Limbova 12, 833 03 Bratislava, Slovakia

**Keywords:** inflammatory bowel disease, IBD, stressors, ulcerative colitis, Crohn’s disease

## Abstract

Background: Stress has been suggested to play a potential role in inflammatory bowel disease (IBD) pathogenesis, but studies focussing on the occurrence of specific life stress events among IBD patients are scarce. Therefore, the aim of the present study was to explore the association between various life stress events and IBD. Methods: Patients with IBD (N = 98, mean age: 38.45, 54.1% men) were compared to a group of healthy controls (N = 405, mean age: 36.45, 58.0% men) originating from a health survey conducted on a representative population sample of Czech adults. The Life Stressor Checklist-Revised (LSC-R) was used to assess the stressors. Results: IBD patients had higher odds of reporting life stressors overall (*p* < 0.001), life stressors before the age of 16 (*p* < 0.004) and a higher score in traumatic stress (*p* < 0.005) and interpersonal violence (*p* < 0.001) when compared to the control group. Gender- and diagnosis-related differences are discussed. Conclusion: Reporting life stressors experienced during childhood or adulthood is strongly associated with IBD. This should be considered in illness management, especially in a severe course of IBD.

## 1. Introduction

Inflammatory bowel disease (IBD) is a group of chronic health conditions with increasing prevalence in the population [1]. Crohn’s disease (CD) and ulcerative colitis (UC) are the representative diseases of IBD, characterised by an inflammatory condition of the gastrointestinal (GI) tract. IBD arises from an interaction between genetic susceptibility and environmental factors [2]. IBD is characterised by relapsing and remitting phases; GI symptoms, such as diarrhoea, nausea, blood in the stool, abdominal pain, sometimes with joints and skin problems and fatigue; and the development of complications, such as intestinal perforations, strictures and fistulas [3]. Moreover, patients with IBD are at higher risk of being diagnosed with an anxiety disorder, depression and a bipolar disorder, which can affect the course of this chronic condition [4,5].

Stress, chronic and acute, is associated with alterations in systemic immune and inflammatory function, which may be relevant in the pathogenesis of IBD [6]. Many studies of the association between stress and IBD confirm the importance of stressful life events [7,8,9]. Patients often notice a connection between life events and a worsening of the illness [10]. However, there is not yet any proof of a direct connection between life events and clinical flares of the disease [11].

Early life events, such as a complicated birth and a lack of breastfeeding, as well as later childhood stressful events are considered to be potential risk factors for IBD [12]. Further, the effect of adverse childhood experiences, such as sexual abuse, physical abuse and witnessing domestic violence, in IBD were part of a study of Fuller-Thomson et al. [13].

Although a strong relationship between perceived stress and GI symptoms has been determined, a study of Targownik et al. [14] showed that perceived stress is unrelated to intestinal inflammation (measured by faecal calprotectin) in CD and only weakly associated with UC. In the study of Wintjens et al. [15], the occurrence of life events and novel perceived stress was associated with disease flares in the following three months, while the presence of perceived stress in general was not. Other prospective studies with clinical indicators are needed to determine whether stress can influence future inflammatory activity and to explore the effectiveness of psychological interventions.

Patients suffering from IBD experience an increased level of stress in relation to work, family and financial pressure [16]. In a Venezuelan study [17], 83% of IBD patients reported stressful situations (stressful economic situation, problems at work, death in the family, divorce, domestic violence, sexual abuse) during the year before the onset of the disease.

Only a few studies of stress in IBD have used a healthy control group [18]. Given this gap in the available literature, the types of life stressors reported by IBD patients need to be examined. The main focus of this study was to explore the association between stressors and IBD. We hypothesised that the group of patients with a severe course of IBD would report a higher occurrence of stressors throughout their life compared to a healthy control group. Another aim of the study was to explore the gender differences in IBD patients compared to the healthy controls and to explore differences in reported stressors between UC and CD patients.

## 2. Methods

### 2.1. Sample

Two groups were compared in this study: (1) A control group originating from a health survey conducted on a representative population sample in the Czech Republic. Data on this sample (*N* = 1800) were collected by trained administrators using face-to-face interviews during September and November 2016. For the purposes of this study, we identified 405 respondents reporting no chronic conditions (i.e., who consider themselves healthy).

(2) A group of IBD patients. These patients were selected at random at two Czech IBD centres (the 2nd Internal Clinic and Surgical Clinic of the Faculty Hospital in Olomouc and Vitkovice Hospital in Ostrava). Collection of the clinical sample took place in May and June 2017. Of the total number of 110 questionnaires, 98 were completed. The whole clinical sample with IBD (*N* = 98, mean age 38.45, 54.1% men) consisted of 62 patients with Crohn’s disease (CD, *N* = 62, mean age 36.18, 53.2% men) and 36 patients with ulcerative colitis (UC, *N* = 36, mean age 41.48, 55.6% men). The IBD patients were asked to fill in the same questionnaire battery as the control group. The patients were approached by healthcare professionals who were trained to assist in completing the questionnaire. The patients completed the questionnaires themselves during a visit to the gastroenterological clinic, assisted by a healthcare worker. The clinical sample describes a “non-average” group of IBD patients, because the patients were recruited at gastroenterological centres which concentrate patients with complicated disease course (93.2% on biological therapy, 6.8% after the surgery).

Respondents agreed to participate in the study by signing an informed consent prior to the study. This study was approved by the ethical committee of the university and conducted in accordance with the protection of personal data (Act. No 101/2000 Coll.).

### 2.2. Measures

#### 2.2.1. Sociodemographic Data

Participants reported gender (male or female), age (continuous), living arrangement (living with a partner in a marriage or a partnership, alone, with parents or siblings), education (primary, completed apprenticeship, secondary school graduated, college or university), and economic activity (disabled, employed, entrepreneur, in household, pensioner, student, unemployed).

#### 2.2.2. Life Stressors

The Life Stressor Checklist-Revised (LSC-R) is a questionnaire that captures traumatic experiences during childhood and during later life. In the Czech Republic, the questionnaire was validated as a clinically useful tool for detecting life stressors and also for research purposes [19]. The LSC-R is a 30-item questionnaire; 19 items describe situations which are subject to the definition of psychological trauma, and 9 describe other stressful life situations. There are 2 open questions with the option of writing other traumatizing events that happened to the respondent or a person close to the respondent. The last part of each item asks how much the stressor affected the life of the person during the last year.

Wolfe et al. [20] recommends either simply adding positively coded items (scores range from 0 to 30) or assessing the impact of stressors in the last years by multiplying each positively coded item by the score of the impact of the stressor (coded from 1 to 5, thus the total score can range from 0 to 150). This method identifies events with traumatic potential (disasters, accidents, robberies, death or sudden death of a family member, abortion in women, abuse and neglect, interpersonal violence). By adding those that made the respondent feel intensive fear or helplessness and fear for life at the time they occurred, we can add up the score of traumatic stress [20,21]. Because of the option to report the age when the stressful event occurred, authors from Mexico [22] recommend assessing the score of life stressors experienced before 16 years of age along with assessing the impact on the last year. Authors from Colombia [23] use the score of interpersonal violence, which contains 10 stressors associated with emotional abuse, physical abuse, sexual abuse and emotional and physical neglect in childhood as well as interpersonal violence in adulthood. In this study, we used several types of scoring detailed above: (1) The summary score of life stressors (LS) with assessment of the impact of stressors in the last year; (2) The score of life stressors experienced before the age of 16 years (LS before 16 y.); (3) A traumatic stress (TS) score containing events with traumatic potential that made the respondent feel intensive fear or helplessness and fear for their life at the time they occurred; (4) The interpersonal violence (IPV) score.

The advantage of the LSC-R is that, in addition to capturing stressors, it measures the personal meaning of stressors in a specific patient. The LSC-R accounts for potential stressors but also captures the participant’s feelings concerning the life event. The LSC-R tool can be one of the options for orientation in a patient’s life. Some items on the questionnaire can be traumatizing for people just to think about, which is why at the end of this study the possibility of a psychotherapeutic intervention was offered.

### 2.3. Statistical Analyses

For describing sociodemographic characteristics, means and standard deviations (SD), frequencies and percentages were used. For all other analyses, including logistic regression, the scores of life stressors in total, the stressors before the age of 16 years, events considered to be traumatic stress events and the events comprising interpersonal violence were dichotomised according to their mean + SD value. The differences in the occurrence of life stressors between the groups were assessed using a test of proportions (Z-test).

Binary logistic regression was used to model the odds of being in the IBD group, depending on the occurrence of life stressors. All the regression models were adjusted for the age and gender of the respondents. A statistical significance level of α = 0.05 was used in all the tests.

All the statistical analyses were performed using the IBM SPSS software, version 21 (IBM, Armonk, NY, USA).

## 3. Results

### 3.1. Sociodemographic Characteristics

The sociodemographic characteristics of the research groups, i.e., the healthy control group (HC) and the IBD patient’s group, are described in Table 1. In a comparison of the sociodemographic characteristics, IBD patients had a significantly higher proportion of married individuals than the HC group. IBD patients also had a higher proportion of disabled and a lower proportion of employed participants compared to the group of HC respondents.

### 3.2. Occurrence of Life Stressors with an Impact on the Last Year in the Research Groups

The occurrence of life stressors in general, life stressors before the age of 16 years, life stressors reaching traumatic stress, and interpersonal violence in the research groups are depicted in Figure 1. The IBD group had a significantly higher occurrence of all types of life stressors compared to the HC group (*p* < 0.05).

The occurrence of individual life stress events in the research groups is depicted in Figure 2. Life stressors with low prevalence (adoption, own imprisonment, serious illness of own child, separation from own child, four questions about forced sexual touching and intercourse before 16 and after 16 years as well as two open questions) are not included in analyses of individual stressors, as their statistical assessment was not possible. The IBD group experienced a significantly higher proportion (*p* < 0.05) of almost all the assessed life stressors, with the exception of experiencing a disaster, accident, financial problem and sudden death of a family member.

Figure 3 and Figure 4 show the occurrence of individual life stressors in the research groups of men and women, respectively. Men with IBD experienced a significantly higher proportion (*p* < 0.05) of separation of parents, serious illness, death in the family, robbery, robbery witness and physical abuse before the age of 16 years compared to healthy men. Women with IBD experienced a significantly higher proportion of having a family member in prison, separation, serious illness, abortion, serious illness and nurturing in a family, sudden death in the family, psychical abuse and neglect, violence in the family and physical abuse after 16 years of age.

### 3.3. Life Stress Events Affecting the Odds of IBD

To assess the odds of being in the IBD group, depending on the occurrence of life stressors, a logistic regression was used. The HC respondents were considered as the reference group. The reporting of life stressors in total, life stressors before the age of 16 years, life stressors reaching traumatic stress and life stressors indicating interpersonal violence was significantly associated with IBD (OR 3.26, 2.82, 2.34, and 3.98, respectively), see Table 2. All individual life stress events were significantly associated with IBD, except for the sudden death of a family member. The highest odds of being in the IBD group was found in respondents experiencing serious illness (OR = 9.2), physical abuse after the age of 16 years (OR = 7.3), physical abuse before the age of 16 years (OR = 4.7) and being a witness to a robbery (OR = 4.5).

In a comparison of the UC and CD patient groups, we found that reporting interpersonal violence, physical abuse before the age of 16 years, sexual harassment and a sudden death in the family was significantly associated with UC (OR = 3.03, 5.63, 4.57 and 3.77) and that indicating a serious illness as a life stressor was significantly associated with CD (OR = 0.26).

## 4. Discussion

The aim of the study was to explore the association between life stressors and IBD. Our main hypothesis—that the group of patients with a severe course of IBD would report a higher occurrence of stressors throughout their life compared to healthy controls—was supported. In our sample, IBD patients had higher odds of reporting life stressors in total, life stressors before the age of 16 years, life stressors reaching traumatic stress, as well as life stressors indicating interpersonal violence. The differences between men and women and UC and CD are discussed below.

In our study, reporting life stress events connected to the family, e.g., one’s own serious illness, serious illness and nurturing in the family, a death in the family, separation of parents, partner separation or the imprisonment of a family member, was strongly associated with IBD. A study by Jakobsen et al. [24] found that the frequency of parental divorce (before the diagnosis of IBD in their child) was higher in IBD patients than in healthy controls. In our study, mostly men with IBD reported a higher occurrence of parental divorce. A significant association between parental stress or family factors and psychological morbidity in young people with IBD was found in a study by Brooks et al. [25]. Family stressors, divorce and loss of family members were found to increase pain-related distress in children with IBD by influencing maladaptive coping and depression symptoms [26]. Greater family stress was associated with a higher occurrence of children’s pain-related expressions of distress and passive coping. A death in the family is a strong stressor mentioned by many authors and practitioners who work on the family approach [27,28,29]. In the present study, women with IBD reported a higher occurrence of nurturing a family member and a sudden death in the family, while a higher proportion of men with IBD reported a death in a family compared to healthy controls.

Overall, suffering from a serious illness was reported to be one of the most common stressors by the IBD patients in our study, mostly in men and in CD patients. IBD itself turns out to be a serious stressor [30]. In particular, patients with active or fluctuating symptoms of IBD rate the illness itself to be a highly frequent source of stress [16]. Therefore, coping with a chronic illness is really important to consider during the treatment of IBD. The study of Pellisier [31] showed that illness perception can be positive or negative—positive emotional adjustment was associated with problem-focused coping, and negative emotional adjustment was associated with emotion-focused coping and external health locus of control. Illness perception has a strong impact on psychological morbidity. Knowing and understanding the perception of illness, coping strategies and the effect on illness outcomes can help health professionals understand the behaviour of IBD patients and improve their quality of life by supporting the ability to cope with stressors related to the disease [32,33]. In a study of young people [34], the emotional impact of IBD (emotional representation) was an independent predictor of all measures of psychological morbidity. The emotional impact of IBD can be associated with a higher level of catastrophizing, a tendency towards pain chronification and with poorer coping skills [35]. There seems to be a vicious circle around IBD: difficulties in coping with the illness cause stress and stress is then a trigger for IBD flares [36]. A recent study confirmed the bidirectional pathways between stress and IBD [37]. Targeted interventions in patients with negative illness perception may be crucial in changing their behavioural patterns towards better self-management. Developing and implementing targeted psychological interventions into the course of IBD treatment is an essential direction for future research and practice. It is important to consider not only the influence of stress on the course of the IBD disease, but to consider IBD as a stressor, as well [18].

A comparison between the UC and CD groups in our study shows that reporting serious illness as a stressor was strongly associated with CD. A German multicentre study of patients with CD and UC [38] found that UC patients in remission were minimally affected by psychological disorders, while CD patients in remission showed insecurity, paranoid ideation and higher neuroticism compared to healthy controls. During the active phase of the disease, both the UC and CD patients scored higher on psychological disorders and maladaptive stress coping tests. More studies assessing the differences between UC and CD are needed to improve target interventions in these chronic conditions.

In our study, IBD was associated with a higher occurrence of all life stressors before the age of 16 compared to the healthy controls. Although only limited research on adverse childhood events and IBD has been conducted, there is some evidence of a relationship between them. Thoman, Lis and Raindl [39] showed that almost half of IBD patients reported some kind of adverse childhood experience, most often neglect. In their study, IBD was also related to stronger anxiety. A clinical study by Drossman et al. [40] revealed that over one-third of UC patients reported a history of physical or sexual abuse. Moreover, patients with a history of abuse referred to a significantly worse overall health status. Self-reported childhood abuse was identified as a significant predictor of poorer health status in adulthood in several studies [41,42,43]. In a study of Caplan et al. [43], childhood abuse and avoidant attachment were significantly associated in patients with UC, but not with CD. The difference between those two groups was explained by a smaller number of patients diagnosed with CD and because of the complex nature of the CD, leading to a lower testing power. Similarly, according to Fuller-Thomson et al. [44], childhood maltreatment (physical and sexual abuse) was associated with UC and not CD. In line with these findings, our study shows that patients with UC have higher odds of reporting interpersonal violence, physical abuse before the age of 16 years and sexual harassment compared to patients with CD.

A population-based study also concluded that IBD is strongly related to a generalised anxiety disorder, especially in women suffering from chronic pain and those with a history of childhood sexual abuse [13]. Differences in reporting stressors between men and women are present in our study as well. From previous studies it is known that self-reported quality of life in IBD patients is lower in females than in males [45]. Our results show that men and women with IBD can differ in their perception of experienced stressors.

Our study assessed life stressors reported by patients with a severe course of IBD. It is important to take into consideration that the stressors were evaluated subjectively and that two people who have experienced the same stress event may experience the situation differently, with a different stress level. Perceived stress is a concept which points out that the degree of stress experienced in a specific person can be influenced by various factors, such as personal characteristics, lifestyle, social support and type of life event [46]. The LSC-R instrument asks about specific stressors but also measures perceived stress and the impact of the severity of these stressors on the person.

The results of the logistic regression in our study show that experiencing adverse childhood and adulthood events is associated with a severe form of IBD. Life stress and abuse history have physiological and behavioural effects that amplify the severity of the condition experienced. These effects lead to increased seeking of health care and explain the higher association of abuse histories with GI illness in referral centres and specialty groups when compared with primary care [47]. Differences in occurrence of life stressors between men and women with IBD and between patients suffering from UC or CD should be considered when creating appropriate therapeutic interventions in gastroenterological practice.

This study provides some valuable information that can help practitioners address their patients’ most frequent stressors. Recalling and talking about difficult life situations can also be a starting point for psychotherapeutic work during treatment of the disease. Patients with IBD often perceive stress as the leading cause of their disease. This can lead to feelings of self-blame, thus causing a relapse. To prevent this vicious circle of stress, gastroenterological practise should be interconnected with mental health practice. Our study indicates that continuous monitoring of IBD patients is very important, because personalised care can help prevent disease flares and the accompanying mental health complications. A recent study recommends asking questions about mental issues, stressors and quality of life in common gastroenterological care [48]. Trying to integrate knowledge about the management of IBD and the presence of a psychologist as a part of the team in the gastroenterological care looks to be a challenge for patient-oriented biopsychosocial care in IBD therapy [49]. The LSC-R questionnaire appears to be appropriate for mapping the situation around potential stressors or traumatization in patients with IBD.

## 5. Strengths and Limitations of the Study

The strength of this study is that only patients with a clinically determined IBD diagnosis coming from IBD centres were involved, i.e., they were not just self-reported. The results may be affected by the fact that patients were receiving biological treatment or have had surgery. This means that these patients had a severe course of the disease and were in a situation when the first choices of conservative cure were not sufficient. On the other hand, the fact that only patients with severe course were involved is in conformity with the need to differentiate the disease severity in subgroups of IBD patients—in our study group those with a severe course of the disease [50]. Another strength of this study is the presence of a control group of healthy people.

A limitation of the study is its cross-sectional character, which does not allow us to assess the causality between stress and exacerbation of the disease. The higher occurrence of stressors in IBD could have been affected by the patients’ severe form of the disease. This means that the results cannot be generalised to all IBD patients. The differences between the severe and mild or average course of the disease should be assessed in future studies. The sample of IBD patients was selected randomly at only two large IBD centres in the Czech Republic, not throughout the Czech Republic. We do not know the onset of the disease in these patients, whether it occurred in childhood or in adulthood.

The healthy control group includes respondents who reported being healthy. Even if some respondents could underreport their health problems, the assessment of health status by self-report is a valid option in national multimorbidity studies [51]. We assume that respondents from the representative Czech sample and from the IBD sample reported health conditions confirmed by a medical diagnosis. The control group included only respondents who reported being without any chronic health conditions.

## 6. Conclusions

Our study shows the connection between a higher occurrence of life stressors and IBD patients with the severe form of the disease. A comparison with healthy controls shows us that patients with severe forms of IBD refer to a higher occurrence of overall life stressors, life stressors before the age 16 years, interpersonal violence and traumatic stress. This should be considered in the management of the illness, especially with severe forms of IBD, when conservative treatment is not sufficient and target psychotherapeutic interventions could be valuable. Possible differences between men and women with IBD and patients with UC and CD in experiencing life stressors should be considered in psychosocial management of the illness as well. The LSC-R is a possible tool for catching stressors and a starting point for communication about the role of adverse life events in a patient’s life.

## Figures and Tables

**Figure 1 ijerph-18-03237-f001:**
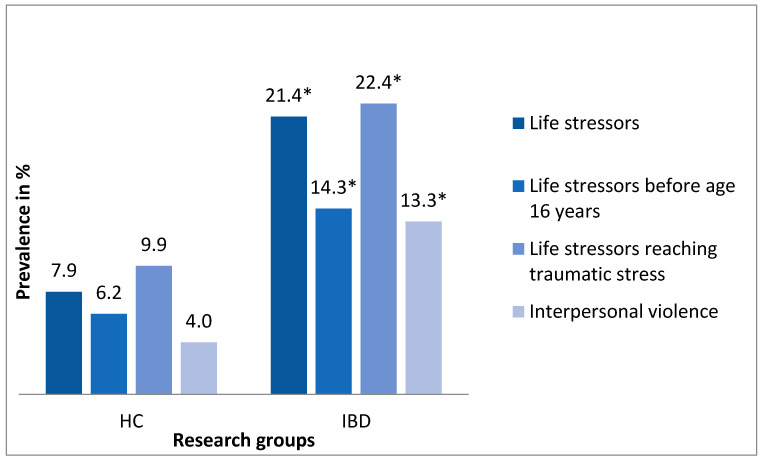
Prevalence of life stressors in the research groups: HC = healthy control group (*N* = 405), IBD = inflammatory bowel disease patients (*N* = 98). The differences in the occurrence of life stressors between the groups were assessed by a test of proportions (Z-test); * *p* < 0.05. Life stressors scores dichotomised according to mean + SD value.

**Figure 2 ijerph-18-03237-f002:**
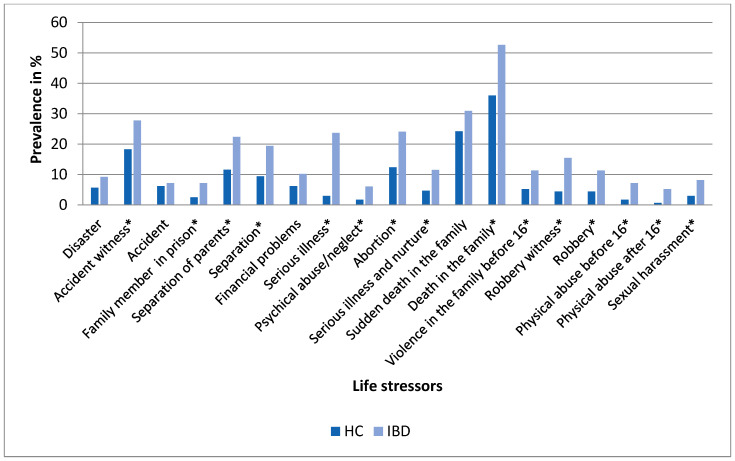
Prevalence of life stressors (%) in the healthy control group (HC, *N* = 405) and in patients with inflammatory bowel disease (IBD, *N* = 98). Differences in the occurrence of individual life stressors between the groups were assessed by a test of proportions (Z-test); * *p* < 0.05. Life stressors with low prevalence are not presented.

**Figure 3 ijerph-18-03237-f003:**
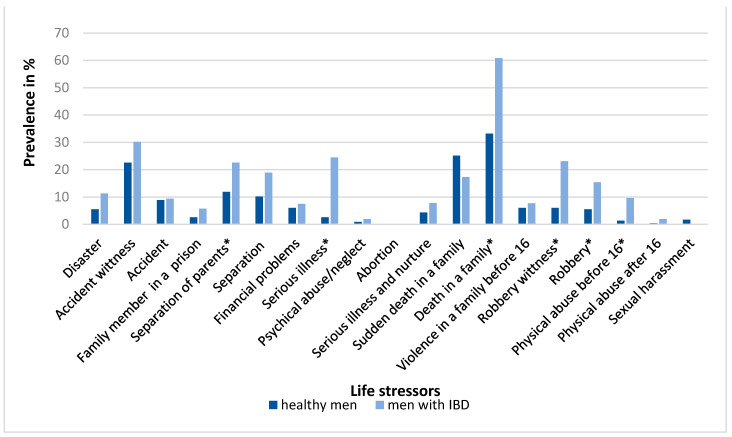
Prevalence of life stressors (%) in the control group of healthy men (*N* = 245) and in men with inflammatory bowel disease (*N* = 53). Differences in the occurrence of individual life stressors between the groups were assessed by a test of proportions (Z-test); * *p* < 0.05.

**Figure 4 ijerph-18-03237-f004:**
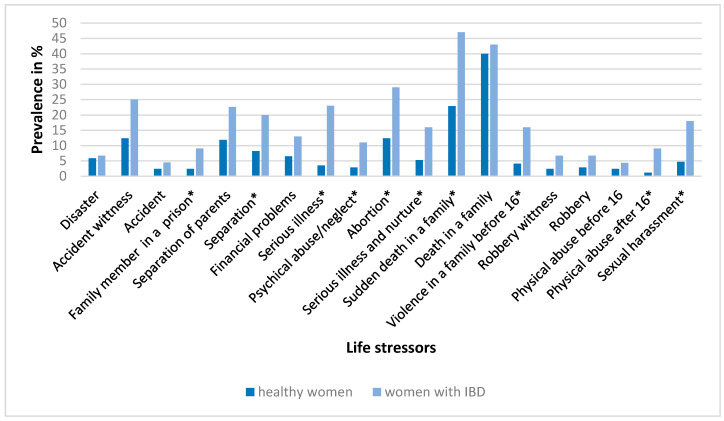
Prevalence of life stressors (%) in the control group of healthy women (*N* = 170) and in women with inflammatory bowel disease (*N* = 45). Differences in the occurrence of individual life stressors between the groups were assessed by a test of proportions (Z-test); * *p* < 0.05.

**Table 1 ijerph-18-03237-t001:** Sociodemographic characteristics: the healthy control group (HC) and the group of patients with inflammatory bowel disease (IBD).

Sociodemographic Group		HC *N* = 405 *N* (%)	IBD *N* = 98 *N* (%)
Age (M ± SD)		36.45 ±14.36	38.45 ± 11.78
(Min–Max)		15–75	19–64
Gender	Men	235 (58.0)	53 (54.1)
	Women	170 (42.0)	45 (45.9)
Living arrangement	Married	152 (37.5)	49 (50.0) *
	With a partner	98 (24.2)	19 (19.4)
	Alone	94 (23.2)	19 (19.4)
	Other (with parents, siblings)	61 (15.1)	11 (11.2)
Education	Primary school	18 (4.4)	2 (2.0)
	Completed Apprenticeship	81 (20.0)	27 (27.6)
	Secondary school	212 (52.3)	42 (42.9)
	College/University	94 (23.2)	27 (27.6)
Economic status	Disabled	2 (0.5)	23 (23.5) *
	Employed	248 (61.2) *	45 (45.9)
	Entrepreneur	49 (12.1)	8 (8.2)
	In household	11 (2.7)	4 (4.1)
	Pensioner	14 (3.5)	4 (4.1)
	Student	71 (17.5)	10 (10.1)
	Unemployed	10 (2.5)	3 (3.1)

Note: Differences in the occurrence of life stressors between groups assessed by a test of proportions (Z-test); * *p* < 0.05.

**Table 2 ijerph-18-03237-t002:** Odds of being in the IBD group, depending on the occurrence of life stressor events. Results of binary logistic regression models with the reference category HC, adjusted for age and gender. In the second column, odds of being in the UC group, depending on the occurrence of life stressors events, with the reference category CD group.

	IBD vs. HC		UC vs. CD	
	OR (95% CI)	*p*-Value	OR (95% CI)	*p*-Value
**Life stressors**				
Life stressors in total	3.26 (1.78–5.99)	0.001	2.27 (0.79–6.48)	0.126
Life stressors before the age 16 years	2.82 (1.39–5.73)	0.004	2.43 (0.69–8.61)	0.168
Life stressors reaching traumatic stress	2.33 (1.29–4.24)	0.005	1.93 (0.68–5.42)	0.214
Interpersonal violence	3.98 (1.81–8.72)	0.001	3.03 (0.86–10.64)	0.084
**Individual life stress events**				
Disaster	1.70 (0.76–3.81)	0.199	0.88 (0.20–3.98)	0.874
Accident witness	1.81 (1.08–3.03)	0.025	1.09 (0.42–2.84)	0.863
Accident	1.04 (0.41–2.64)	0.932	0.82 (0.13–5.17)	0.837
Family member in prison	3.14 (1.16–8.49)	0.025	0.88 (0.16–5.02)	0.888
Separation of parents	2.43 (1.37–30)	0.002	0.96 (0.33–2.79)	0.943
Separation	2.15 (1.13–4.10)	0.020	0.48 (0.14–1.61)	0.237
Financial problems	1.70 (0.78–3.68)	0.181	0.37 (0.69–1.94)	0.238
Serious illness	9.19 (4.32–19.54)	0.001	0.26 (0.76–0.919)	0.036
Psychical abuse and neglect	3.94 (1.27–12.21)	0.018	2.61 (0.46–14.86)	0.280
Abortion *	3.07 (1.31–7.19)	0.010	1.59 (0.38–6.73)	0.526
Serious illness and nurturing in the family	2.55 (1.15–5.63)	0.021	0.87 (0.22–3.43)	0.847
Sudden death in the family	1.43 (0.87–2.34)	0.157	3.77 (1.33–10.67)	0.012
Death in the family	1.99 (1.25–3.14)	0.003	1.61 (0.66–3.95)	0.296
Violence in the family before the age 16 years	2.51 (1.16–5.44)	0.019	1.40 (0.36–5.51)	0.629
Robbery witness	4.45 (2.12–9.34)	0.001	0.89 (0.26–3.08)	0.854
Robbery	2.34 (1.01–5.44)	0.048	1.89 (0.43–8.31)	0.397
Physical abuse before the age of 16 years	4.69 (1.60–13.76)	0.005	5.63 (0.97–32.23)	0.054
Physical abuse after the age of 16 years	7.32 (1.70–31.49)	0.008	3.09 (0.46–20.97)	0.247
Sexual harassment	2.97 (1.16–7.64)	0.024	4.57 (0.87–24.03)	0.072

Note: OR = odds ratio; IBD = Inflammatory Bowel Disease group; UC = ulcerative colitis group; HC = healthy control group. * Abortion was assessed separately only in women.

## Data Availability

Data for this manuscript may be made available by request.

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
