# Peer review of "Life Stressors in Patients with Inflammatory Bowel Disease: Comparison with a Population-Based Healthy Control Group in the Czech Republic"

_ijerph, 2021, doi:10.3390/ijerph18063237_

Round 1

Reviewer 1 Report

General

The article deals with the topic of life stressors including early life stressors in the teen hood, in inflammatory bowel disease in comparison with a healthy control group in the Czech Republic. The topic covered in this article is of primary interest in the field of IBD and to date few researchers have addressed the issue of early stressors in this population. One of the strong points of this study is the existence of a control group of healthy subjects of substantial size. The methodology is of good quality and the discussion well conducted even if could be improved regarding some points.

I have few major remarks to make regarding the hypothesis, the way results are presented regarding this main hypothesis and some conceptual approaches in the discussion.

Introduction:

At the end of the introduction, lines 78-80 please include the hypothesis regarding the control group and not only that regarding the link between stressors and IBD. Why did the authors conducted a control group if no hypothesis are formalised in relation with this group?

Results:

The main hypothesis announced in the introduction is that “patients of severe course of IBD would report a higher occurrence of stressors throughout their life”. However, this main hypothesis is neither announced in the abstract and nor addressed as such in the results. This is not really clear how the authors assessed the severity of IBD and perform the link between the prevalence of life stressors and the severity of IBD? This issue must be addressed.

Is there any differences concerning life stressors prevalence between UC and CD? If not, this must be precise.

Discussion:

Paragraph 4.2: line 216: please give more details about the way coping is influenced by stressors

Paragraph 4.3: line 228: the illness perception must be conceptually defined, previous studies supported this idea of a positive vs negative emotional adjustment regarding the illness in IBD related to different coping style PMID: 19910123.

Paragraph 4.4: line 263: please discuss the question of the overlap between IBD and IBS

Paragraph 4.5: line 280-282: the concept of perceived stress must be introduced here and discussed.

The conclusions of the study must be in line with the different hypothesis. This can be easily improved.

Reviewer 2 Report

In this study, authors explored the occurrence of the life stressors in IBD patients with complicated disease course with comparison to healthy controls.

The general idea of this article is very interesting, and it provides some interesting data regarding this topic as well. However, some aspects of the study should be addressed, and key questions answered before potential publication.

First, there is a need to make a thorough check-up of English language. There are many grammatical errors, and in some parts of the article, exact meaning (message) of the sentences and phrases is not easy to understand.

As it seems, this article provides novelty to the field. However, introduction section should be re-organized. There is no structured flow of the information, and concise overview of the studies that are connected to the topic should be provided. Many studies are mentioned in various parts of introduction, but it feels without order. Lines 43-68 should be re-written in that direction.

Are there any studies in literature that explored stressful life events and IBD in any way? If there are, they should be mentioned in introduction.

Line 46 – it says that patients often notice connection between life events and symptoms of IBD – is there reference for this statement?

Line 47 – This is one of most important introduction sentences, as it mentions gap in literature. It should be written near the end of section

Lines 69-77 – this part of the introduction is not very important or connected to current topic. Moreover, it is written at the end of the introduction, where general information for the need of this study should be provided, and final aim explained.

Line 78 – I’m not sure that phrase “need to map the situation in Czech Republic…” is convenient. What exactly is this need? This study should be applicable to IBD population in general, therefore I suggest more general aim construction

Line 87 – it seems that control group was elected solely on their subjective report – this should be addressed in limitations section

Line 102-103 – this could be moved to limitations as well

2.2.2. Life stressors – more information regarding this questionnaire could be provided, even though references are placed – as authors state, they used 4 different scoring types – how many items they consist of? What are these items? Can this questionnaire be quantified?

Table 1 – I find that second column can be deleted, as that information can be placed in footnotes or other places in table

Figure 1 – it is unclear of how many items these categories consist of, and if all 30 items are used?

Furthermore, in Figure 2 only 19 stressors are mentioned. Are other items analysed? They are mentioned in the methods (9 other stressful life situations, and 2 open questions).

More results could be explored, as potential differences between the gender – some interesting information could be found there, so I suggest additional analyses

Table 2 – 95% confidence intervals should be added in the table

Discussion section is quite long. I suggest removing of subsections and concise re-arrangement of this section with only most important information that is related to article topic. Furthermore, 4.1 subsection is more appropriate to be put in introduction

Lines 291-294 – this section is more appropriate to be put in methods section

Round 2

Reviewer 2 Report

The authors have been responsive, and overall presentation of the manuscript improved significantly.

I have only one further remark, and it involves LSC-R questionairre. I find that additional, more detailed expansion of the exact scoring system is still needed. Moreover, if some of the items of the 30-item scale were not presented in the results, it should be clearly mentioned in the text.

Also, in statistical analysis section, it is stated that mentioned scores were dichotomised for the purpose of logistic regression, and Figure 1 is not addressing that.
